# Graph Heavy Light Decomposed Networks: Towards learning scalable long-range graph patterns

**Alan Marko**[*]
University of Cambridge
am2677@cantab.ac.uk

**Peter D. Ralbovsky**[*]
University of Cambridge
pdr34@cantab.ac.uk

## Abstract

We present graph heavy light decomposed networks (GraphHLDNs), novel neural network architectures allowing reasoning about long-range relationships on graphs reducible to trees. By decomposing the trees into a set of interconnected chains in a way similar to the heavy-light decomposition algorithm, we rewire a tree with $n$ vertices so that its depth is in order of $O(\log^2 n)$ after building a binary-tree-shaped neural network over each chain. This enables faster propagation and aggregation of information over the whole graph while being able to reason about long-range sequences of nodes considering their ordering. We show that this method partially addresses the previous need for message-passing architectures for step-by-step supervision to execute certain algorithms out-of-distribution. Our method is also applicable to real-world datasets, achieving results competitive with other state-of-the-art architectures targeted at learning long-range dependencies or using positional encodings on several molecular datasets.

## 1 Introduction

In most graph neural network architectures where in each layer nodes aggregate information from their neighbours, the range in which the information can travel is limited by the number of propagation layers. This hinders the ability of such architectures to reason about long-range dependencies, patterns and metrics such as orderings of vertices, their distances, or attributes of paths between two or more nodes.

Furthermore, even if the network manages to learn and recognise such patterns on smaller graphs (i.e. by using a step-by-step supervision signal as in [1]), the networks have poor ability to generalise such patterns out-of distribution to graphs of larger scales and sizes [1, 2].

Several recent works tried to tackle long-range reasoning. Approaches include addition of various positional encodings [3–5], hierarchical networks that make connections between distant nodes [6] or inclusion of modules that dynamically change the number of propagation layers based on the task or graph size [2]. However, these methods have limitations: For example, hierarchical networks merge multiple nodes together, which leads to a loss of information about their original edge connections. On the other hand, modules dynamically changing the number of propagation layers usually require a linear number of steps depending on the graph diameter leading to over-smoothing and diminishing/exploding gradient problems on large graphs.

In this work, we propose a novel architecture that allows better reasoning over long-range distances on trees and graphs easily reducible to trees – mainly ones where the difference between the number of vertices and edges is small. This is done by first reducing the graph to a tree – for example by taking its spanning tree, then decomposing the tree into a set of chains, similarly to the heavy-light decomposition algorithm (introduced by Sleator and Tarjan [7]) and connecting different chains through binary-tree-shaped neural networks. This design allows the network to reason not only about

---

[*]Equal contribution.

Marko et al., Graph Heavy Light Decomposed Networks: Towards learning scalable long-range graph patterns (Extended Abstract). Presented at the First Learning on Graphs Conference (LoG 2022), Virtual Event, December 9–12, 2022.

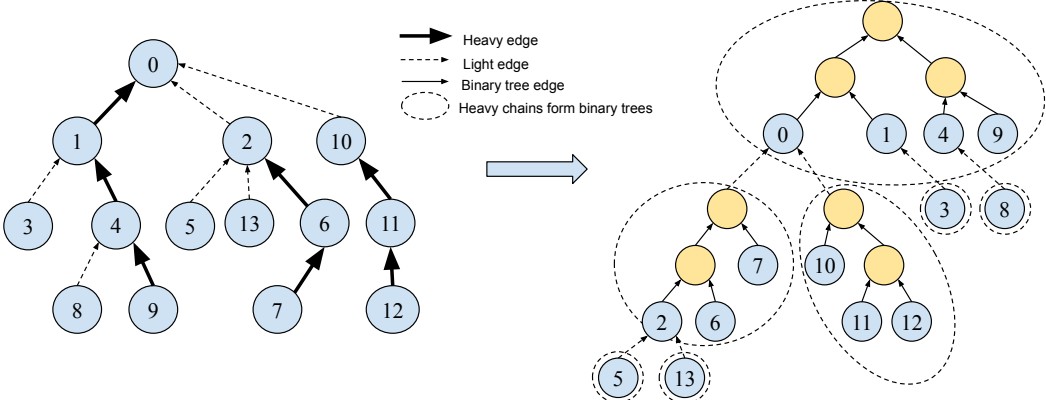

**Figure 1:** In the left part is an example of an input tree rooted with edges split into heavy and light edges. In the right part, each heavy chain was transformed into a binary tree. Binary trees were then connected along light edges. To compute the graph level feature, we use encode, process, decode method, when first all inputs are encoded. In the process phase, the nodes are evaluated bottom-up layer by layer. In binary tree internal nodes (yellow) binary merging MLP $\phi$ is used.

neighbouring relationships but also about larger units such as paths. We show that in this way, our model is able to learn, execute and strongly generalise without step-by-step supervision signal new types of long-range patterns and algorithms that were not possible before, such as finding the shortest path, the lowest common ancestor or minimum vertex cover. Further, we show that GraphHLDN has strong utility on real-world datasets and is competitive or even outperforms best models on molecular datasets such as AQSOL [4], ESOL [8], or `Peptides-struct` [9].

## 2 Methodology

The key step in our method is the generation of the *heavy-light decomposed (HLD)* tree. This consists of three main sub-steps: first selecting light edges splitting the tree into chains, then creating binary trees over each chain and finally connecting those trees along the light edges. Firstly, after rooting the tree, the edges are split into heavy and light ones in a similar way as in the heavy light decomposition algorithm: i.e. so that each vertex has at most one heavy child and from each vertex, the path to the root contains at most $O(\log n)^2$ light edges. After creating the binary trees, we connect them along the light edges. The binary tree root of each chain connected with a light edge in the original tree will be connected with the light edge parent in the original tree as displayed in Figure 1. Please see Appendix A for formal details about how the tree is generated.

To process the input tree we closely follow the setup with encode-process-decode[10] architecture as used in [1] and [11], where GraphHLDN is at the heart of the processing phase. To describe the progression of information through the GraphHLDN we split the nodes into two categories – *merging* nodes (new ones in GraphHLDN tree coloured yellow in Figure 1) and the *original* nodes. To compute graph-level output we traverse through the graph in layers determined by the depths of the individual vertices from the deepest vertices to the root. In each layer, we combine aggregated information from deeper layers to obtain aggregated information for use at higher levels. In each merging node, the representation of two of its children $x_l$ and $x_r$ is combined using trainable multi-layer perceptron (MLP) $\phi$ as $\phi(x_l, x_r)$. [3] In some of the original nodes, we need to process light children as well. In order to do this, we take a representation of each light-children. Then we process them through separate MLP $\phi_2$, and afterwards combine them using sum aggregation. Then MLP $\phi_3$ is used which combines the representation of the original node $x_p$ and element-wise sum $x_c$ obtained from its light-edge children as $\phi_3(x_p, x_c)$. Using these rules layer by layer from the bottom up, the network gradually aggregates the information to the root.

---

[2] $n$ denotes the number of vertices in tree

[3] We can also note that each merging node merges two parts of a heavy chain along some edge in the original tree. Therefore in this merging process, we can, besides encoded children representation, also include the edge representation of this edge.

**Table 1:** Results comparing MPNN with GraphHLDN on synthetic tasks. Each task has exact solution, so average test accuracy is reported in and out of distribution.

| Task description | GraphHLDN | | MPNN | |
|---|---|---|---|---|
| | $n \leq 100$ | $n \leq 10000$ | $n \leq 100$ | $n \leq 10000$ |
| Predict nodes on shortest path | **100%** | **99.95%** | 71.13% | 81.89% |
| Find LCA of 2 nodes with given root | **99.5%** | **91.4%** | 32.16% | 20.68% |
| Predict nodes in MVC [5] | **99.37%** | **99.55%** | 91.27% | 91.02% |

When the goal is to compute node-level targets, we can send the information downwards, traversing in a top-down approach where in each layer, we combine the representation of each node from the bottom-up approach and the representation of its parent from the top-down approach. These aggregation (bottom-up) and spreading (top-down) passes of information can be combined multiple times in order to allow more general functions to be learned.

The choice of this method is beneficial for two reasons. Firstly it is very similar to segment trees which are often used with heavy light decomposition for answering queries about trees in $O(\log^2 n)$ time. This allows it to do the computation using $O(\log^2 n)$ message passing iterations resolving the vanishing gradient problem and also improving generalisation out of distribution because multiplying the number of nodes results in only a constant increase in the number of iterations. Secondly, this structure allows the preservation of ordering information of vertices along the path in the process, as the learnable merging function has left and right vertex as separate and distinguishable inputs. Note that in all layers, we use the same perceptron for merging. Thus, the merging function should work on all different sizes of segments (it should be able to merge vertex representing one node with vertex representing 1024 nodes).

**Associativity consistency loss (ACL).** One of the features which we expect from perceptron merging nodes is therefore associativity. We enforce this by adding ACL, which is computed by taking random triplets of nodes from the tree with their representations $a, b, c$ and then enforcing that $|\mathrm{BN}(\phi(\phi(a,b),c)) - \mathrm{BN}(\phi(a,\phi(b,c)))|$ is minimal. Batch normalisation function BN is used in order to normalise among the features. To make the effect of the normalisation stronger, instead of creating just one heavy-light tree from a defined root, we choose multiple random roots with different corresponding HLD trees, and during testing, we average the output of each such tree.

## 3 Evaluation

We evaluate the proposed architecture on both synthetic algorithmic datasets and molecular benchmarks. In algorithmic datasets the input graphs consist of uniformly randomly selected trees[4] with the training and validation datasets having up to 100 nodes and test sets having up to 10000 nodes to test out-of-distribution generalisation to larger graphs. The evaluation focuses on node classification tasks: prediction of nodes on the shortest path between two marked nodes, finding the lowest common ancestor for two randomly selected nodes and a randomly marked root, and prediction of nodes in the minimum vertex cover. We use a GraphHLDN network with hidden embeddings having size 64 and three-layer multi-layer perceptrons with LeakyReLUs. For comparison, we train a 30 iteration message passing neural network (MPNN) having sum aggregation and the same hidden embedding size and multi-layer perceptrons on full graphs instead of spanning trees.

We compare the performance of GraphHLDN on `Peptides-Struct`, AQSOL, and ESOL benchmarking datasets with previously reported baseline results from [4] [9] and [12]. The only difference is that in the case of `Peptides-Struct` we use hidden embeddings of size 128. For each graph in the datasets, we select a random spanning tree of the graph. If the graph has multiple components, we randomly select just one. We then create 30 randomly chosen transformed HLD trees from the selected spanning trees and, during testing, report how the averaged prediction on all 30 trees compares with the targets.

---

[4]As all synthetic datasets consist of trees, we do not need to erase any edges in this case.

[5]Weighted Minimum vertex cover; if there are conflicts we prefer solutions where selected nodes are as close to root of HLD as possible, this leads to unique solutions. Weights are integers between 1 and 5.

**Table 2:** Results comparing test MAE on AQSOL dataset. The suffix LapPE denotes the use of Laplacian Eigenvectors as node positional encodings with dimension 4.

| Model | L | #Params | Test MAE $\pm$ s.d. |
|---|---|---|---|
| RingGNN | 2 | 123k | $3.769 \pm 1.012$ |
| GIN | 16 | 514k | $1.962 \pm 0.058$ |
| MoNet | 16 | 507k | $1.501 \pm 0.056$ |
| GAT | 16 | 540k | $1.403 \pm 0.008$ |
| GCN | 16 | 511k | $1.333 \pm 0.013$ |
| GatedGCN | 16 | 507k | $1.308 \pm 0.013$ |
| 3WLGNN | 3 | 525k | $1.108 \pm 0.036$ |
| GatedGCN-LapPE | 16 | 507k | $0.996 \pm 0.008$ |
| GraphHLDN | N/A | 87k | $\mathbf{0.882 \pm 0.012}$ |

**Table 3:** Results comparing test MAE on `Peptides-struct` dataset.

| Model | L | #Params | Test MAE $\pm$ s.d. |
|---|---|---|---|
| GINE | 5 | 547k | $0.354 \pm 0.0045$ |
| GCN | 5 | 508k | $0.349 \pm 0.0013$ |
| GatedGCN | 5 | 509k | $0.342 \pm 0.0013$ |
| GatedGCN+RWSE | 5 | 506k | $0.335 \pm 0.0006$ |
| SAN+LapPE | 4 | 493k | $0.268 \pm 0.0043$ |
| SAN+RWSE | 4 | 500k | $0.254 \pm 0.0012$ |
| Transformer+LapPE | 4 | 488k | $0.252 \pm 0.0016$ |
| GraphHLDN | N/A | 351k | $0.288 \pm 0.0032$ |

**Discussion and conclusions.** As displayed in the table 1, GraphHLDN is able to learn the algorithmic patterns from synthetic tasks and generalises out of training distribution to graphs with a hundred times more nodes. This is despite the fact that no step-by-step supervision signal was used to learn intermediate algorithmic steps as required by previous works that could only generalise to much smaller graphs. For most tasks, the precision is near perfect in the case of GraphHLDN, suggesting that the network learns the actual algorithm behind the dataset target rather than some kind of its approximation.

Due to the tree-shaped structure of GraphHLDN, the nodes in each layer need to aggregate and summarise information from nearly twice as many nodes from a deeper layer. This introduces the bottleneck causing over-squashing of exponentially growing information into fixed-size vectors, which was shown to negatively impact the performance of graph neural networks [13] on graphs with negatively curved edges [14]. However, as can be seen in the tables 2, 3 and 4, our empirical evaluation shows that GraphHLDN is not only applicable to synthetic tasks, but it can also be practically useful on molecular datasets. GraphHLDN outperforms all models reported in [4] on the AQSOL dataset while using a significantly smaller number of parameters. Similarly, on ESOL it almost matches the performance of D-MPNN, and in the case of the `Peptides-struct` dataset focused on long-range dependencies, GraphHLDN is competitive with transformer-based architectures.

It is also notable that this performance is achieved despite ignoring certain edges not included in the spanning trees when the input graphs are not trees. We hope that our work will inspire further

**Table 4:** Results comparing test RMSE on ESOL dataset.

| Model | L | #Params | Test RMSE $\pm$ s.d. |
|---|---|---|---|
| Fingerprint + MLP | 5 | 401k | $0.922 \pm 0.017$ |
| GIN | 5 | 626k | $0.665 \pm 0.026$ |
| GAT | 5 | 671k | $0.654 \pm 0.028$ |
| D-MPNN | 5 | 100k | $0.635 \pm 0.027$ |
| GraphHLDN | N/A | 87k | $0.639 \pm 0.019$ |

research in extending the capabilities of GraphHLDN to other graph topologies and further enhancing or combining capabilities of classical message-passing with GraphHLDN.

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

# A    Algorithm for the construction of GraphHLDN tree

In this appendix, we formally describe the algorithm for the construction of the tree structure used by GraphHLDNs. As mentioned earlier, GraphHLDN can be applied to any graph-structured data that are easily reducible to trees. This can be either in the form of direct mapping from a particular graph to a tree or by choosing a subset of edges from a graph that forms a spanning tree of the original graph. For example, as we have shown in the evaluation, in many molecular datasets, the difference between the average number of edges and the average number of vertices is small (typically less than 3)[6], and so we are able to achieve competitive results even despite not utilising the full graph.

The algorithm for the construction of the tree used by GraphHLDN consists of the following three steps:

1. For an input graph $G = (V, E)$ that is not a tree, choose a random spanning tree $T$ from $G$. In this work, this is done by a DFS traversal from a uniformly randomly selected root vertex $v \in V$. The traversal always selects uniformly randomly the next vertex to explore from the available options. In the rare case that the graph has more than one component, we just focus on the component with the largest number of vertices.

2. From tree $T$, the algorithm uniformly randomly selects the root $r \in V$ and roots the tree in this node. To get better results, we can select multiple such roots, compute the result of GraphHLDN for each and then average the results to get the final value.

3. We perform the Heavy Light Decomposition algorithm (HLD) [7] to split the tree into a set of chains. The nodes inside of a single chain are connected by so-called *heavy edges*. The remaining edges are called *light edges* and connect the nodes between different chains as illustrated in Figure 1. This split achieves the property that for any node $v \in V$, the path between $v$ and root $r$ contains $O(\log n)$ light edges and, therefore $O(\log n)$ different chains.

   The HLD algorithm consists of these two steps:

   (a) For each vertex $v \in V$, count the number of nodes in the subtree rooted in node $v$ of the rooted tree $T$. For node $v$, this is denoted as $subtree\_size(v)$.

   (b) For each vertex $v \in V$ that is not a leaf and thus has at least one descendant, select a vertex $u$ from its direct descendants for which the $subtree\_size(u)$ is the largest. Let edge $(u, v)$ be a heavy edge.

   (c) All other edges that are not heavy are light.

   Each light edge $(u, v)$, where $v$ is closer to the root $r$, connects a subtree rooted in $u$ to the remaining graph with at least the same number of vertices. Therefore it can be easily proven that there are at most $O(\log n)$ light edges on any path to the root.

4. Now we transform the rooted tree $T$ with marked heavy and light edges into the final tree used by GraphHLDN as shown on the right side of Figure 1. For every chain of nodes connected by heavy edges, we construct a binary tree whose leaves represent the original nodes of the chain. The binary tree is constructed similarly to the Quick Sort algorithm:

   (a) If chain $c$ has just one node $v_1$, the resulting binary tree will also have just one node corresponding to the original node $v_1$

   (b) Otherwise, for a chain $c$ having nodes $c = v_1, v_2, ..., v_n$ sorted in this order based on how far they are from the root $r$, we select uniformly randomly a node $v_a$ where we split the chain in two halves $- c_{left} = v1, .., v_a$ and $c_{right} = v_{a+1}, .., v_n$.

   (c) We create a new *merging* node $m$ and make its left and right child nodes the roots of binary trees recursively constructed by this process for chains $c_{left}$ and $c_{right}$.

   Since the Quick Sort algorithm can be performed in asymptotically $O(\log n)$ layers, the newly created binary tree also has asymptotic height $O(\log n)$. After all chains are converted to binary trees, the light edges connecting two different chains in the original tree will be replaced by a new edge. This edge $(u, v)$, where $v$ is closer to the root, connects the new node corresponding to $v$ with the root of the binary tree where $u$ belongs.

Since there are $O(\log n)$ chains on any path to the root and each chain was converted to a binary tree that also has depth in the order of $O(\log n)$, the depth of the final tree is $O(\log^2 n)$.

---

[6]indicating that we need to remove at less than 4 edges to obtain a tree

As described in Section 2, in the case of predicting a global property of the graph, we traverse the graph upward, combining information from children into parent nodes. If we want to instead compute node-level targets, we first also traverse the graph in the same way upward and then go downward back to the leaves combining the representation of each node with its parent. The upward and downward passes can be performed multiple times to learn more general functions.

## A.1 Benefits of this design

This design is beneficial for two main reasons outlined in the methodology – scalability and preservation of ordering information.

**Scalability.** Compared to classical message-passing architectures, on many algorithmic reasoning tasks, our model is able to achieve much better out-of-distribution generalisation to graphs with a larger number of vertices than what it was trained on.[7] This is because, in classical message passing, we need as many layers as the length of the path between two nodes between which we want to propagate the information. If the model learns a property of a certain path, it is difficult to generalise this model to longer paths. This can be attributed to a small error introduced by every layer, which grows exponentially with the execution of more layers, and so we quickly encounter the problem of exploding errors or even exploding gradients harming the predictions. However, in our model, if we multiply the length of the paths, the number of layers needed to be executed increases just by a constant, so the errors do not compound exponentially.

**Preservation of ordering.** The second main advantage is that our model preserves the ordering information of vertices along the path. Compared to other hierarchical methods where aggregation of information from multiple nodes happens, our model can distinguish between the information aggregated from the left and right sons. In this way, the model can reason about whether certain features or properties of the path are ordered in a particular way, instead of aggregating them all together. This enables scalable reasoning about new types of long-range patterns that were not possible to model before.

---

[7]In other words: We achieve very good results by evaluating the model on much larger graphs than it was trained on.

