# OpenReview forum: "Graph Heavy Light Decomposed Networks: Towards learning scalable long-range graph patterns"
_logconference.io/LOG/2022/Conference — LoG 2022 Poster_

### Official Review · Reviewer_FX7i · 2022-10-07

**Overall Score:** 5
**Confidence:** 4

**Review:**

Summary:
This paper introduces graph heavy light decomposed networks (GraphHLDNs) to allow reasoning about long-range relationships on graphs reducible to trees. The experimental results show that the proposed method could outperform SOTA methods on several datasets.


Strengths:
1. The idea of using heavy-light decomposition is interesting.
2. The evaluation results show that the proposed GraphHLDNs could outperform SOTA methods on the synthetic and the AQSOL dataset.

Weaknesses:
1. It is not quite clear how to generate the heavy light decomposed tree. More details should be included.
2. The experimental results on the Peptides-struct and ESOL datasets show that GraphHLDNs perform worse than SOTA methods.
3. The proposed GraphHLDNs is only applicable to graphs reducible to trees.

Questions:
1. What are the definitions of light and heavy edges?
2. How to generate the heavy light decomposed tree?

---

### Official Review · Reviewer_QuNj · 2022-10-18

**Overall Score:** 6
**Confidence:** 4

**Review:**

## Summary

This extended abstract presents a message passing framework by constructing a binary tree hierarchy from a general graph. The method first extracts a spanning tree from the graph and then convert the tree into a binary one using the "heavy-light decomposition" algorithm. With the tree structure defined, neural network modules perform bottom-up and top-down message passings following the child-parent relationships. The binary tree structure preserves the ordering (e.g., by differentiating left/right children) and thus is claimed to achieve better performance compared with MPNN on some graph algorithms.


## Strengths

+ The proposed model seems original. There exist works that treat graphs as trees and perform message passing accordingly. Yet I haven't seen existing works converting general graphs into binary trees and then performing the specific message passing operations.
+ The proposed model is evaluated on different benchmarks covering a variety of graph algorithms.
+ Overall, the paper is easy to read.

## Weaknesses

- While the algorithmic steps are clear, the model design lacks motivation and intuitive explanation (see details below).
- The evaluation may not be fair since the authors explicitly drops edges from the original graph and thus creates a somehow "artificial" setup. In addition, since the baselines are not specifically designed for trees, it may be more fair to execute the baselines on the original graph.
- There lacks some important background information. For example, the description of the heavy-light decomposition is too brief and sounds vague.

## Details and recommendation

I have some major concerns regarding the motivation / intuition of the overall design:
* The most fundamental question is: why do we want to convert the graph into a binary tree? Such a process will drop many edges in the original graph and is highly stochastic. More importantly, many mentioned/evaluated graph algorithms (e.g., shortest path, vertex cut, etc) operate directly on graphs instead of their spanning trees.
* On the binary tree, message passing from the leaves to the root takes log^2 n iterations. However, on the original graph, it only takes D message passing iterations to pass messages of all nodes to the target node, where D is the diameter of the graph. D stays small even for massive power-law graphs.
* It is mentioned that one benefit of the design is its preservation of "ordering". It is unclear why we need to preserve such "ordering" in the first place. Many graph properties / algorithms are permutation-invariant and thus the ordering does not matter. In addition, it is unclear if the proposed module can indeed preserve ordering. For example, for right subfigure of Fig 1, if we swap the position of 1 and 0 , I believe it is still a valid heavy-light decomposition. However, the left and right inputs of the MLP aggregator are then swapped, making the ordering not preserved.
* If you randomly construct spanning trees and randomly pick root nodes of the binary tree, it seems impossible to preserve a consistent ordering.

In addition, there are some concerns on the evaluation:
* Since the baselines are designed for general graphs, I think it is better to execute them on the original graph rather than the randomly constructed binary trees.
* Having the size of the test set much larger than the training set does not mean "out-of-distribution" generalization. A large test set can still follow the same distribution as the small training set.

Considering the above, I vote for rejection of the paper.


## Post-rebuttal response

I think the responses have clarified most of my concerns. I have increased my score, hoping that the authors can integrate their response to their next revision of the paper.

---

### Official Review · Reviewer_o3rJ · 2022-10-21

**Overall Score:** 5
**Confidence:** 4

**Review:**

Summary: This paper presents graph heavy light decomposed networks (GraphHLDNs) for reasoning about long-range relationships on graphs reducible to trees. ########################################################################## Reasons for score: Overall, I vote for weakly reject. I think the motivation of this article is not clear, and it also confuses me, because the method turns the constructed graph into a tree, and then its learning process is very similar to methods such as random forest. I hope that the author can clearly explain the motivation more clearly and the advantages of this method in subsequent revisions.

########################################################################## Comments I would like addressed during the rebuttal period: ########################################################################## Cons: 1.	There are many technical issues that exist. In Methodology, how the trees are rooting? how the edges are split into heavy and light ones? What is the heavy light decomposition algorithm? 2.	Why the graphs are transformed into tree structure? In such cases, methods such as random forest should also be used as comparison methods. 3.	It is suggested that the author formalize the representation in the method. How the nodes of the tree split into two categories: merging and original nodes. 4.	Its graph learning process is familiar with the GIN, so how could the proposed method obtain such a large improvement in the experiments? 5.	From the current experiments, it is difficult to explain why the method proposed by the author can achieve such good performance, and the author should conduct more experimental analysis. 6.	There are some writing errors that exist such as “the the root contains”.

---

### Meta-Review · Area_Chair_1TRy · 2022-11-11

**Confidence:** 4
**Recommendation:** Borderline and needs further discussi…

**Meta Review:**

This is a short paper arguing for a permutation-sensitive data augmentation process (e.g., based on DFS) that creates a binary tree from the graph, which then significantly reduces computation effort of the representation (which is now specialized). It is undesirable to have a graph method that is permutation sensitive.

Reviewers agreed the concept is interesting although its application is limited to graphs that area nearly trees (e.g., molecules and some algorithmic reasoning tasks).  The authors did a good job at the rebuttal, improving the paper and clarifying some confusing parts of the paper.

This is a borderline case. Narrow application, a method that is permutation sensitive, and an OOD claim that is not formally defined.


PS: In general, in order to claim OOD generalization to size, one should theoretically define it as a causal intervention on the graph sizes, and then explain why the proposed approach is robust to these interventions. The authors should change the introduction to say it is just an empirical observation in a specific application. It is not a general approach.

---

### Decision · Program_Chairs · 2022-11-23

Accept (Poster)